# Multivariate Analysis of Agronomic Traits in Newly Developed Maize Hybrids Grown under Different Agro-Environments

**DOI:** 10.3390/plants11091187

**Published:** 2022-04-28

**Authors:** Mohamed Omar, Hassan A. Rabie, Saber A. Mowafi, Hisham T. Othman, Diaa Abd El-Moneim, Khadiga Alharbi, Elsayed Mansour, Mohamed M. A. Ali

**Affiliations:** 1Makien hybrids, Beni-Suef 62729, Egypt; m.omaralsaqa@gmail.com; 2Department of Crop Science, Faculty of Agriculture, Zagazig University, Zagazig 44519, Egypt; hassan_rabie@outlook.sa (H.A.R.); saber.mowafi@outlook.sa (S.A.M.); 3Middle Egypt Company, Beni-Suef 62619, Egypt; h.tawfik@middleeg.com; 4Department of Plant Production (Genetic Branch), Faculty of Environmental Agricultural Sciences, Arish University, El-Arish 45511, Egypt; dabdelmoniem@aru.edu.eg; 5Department of Biology, College of Science, Princess Nourah bint Abdulrahman University, P.O. Box 84428, Riyadh 11671, Saudi Arabia

**Keywords:** sowing date, planting density, cluster analysis, stability, joint regression, AMMI, GGE, biplot

## Abstract

Developing high-yielding maize hybrids is essential under the fast-growing global population and abrupt global climate change. Planting density is one of the imperative components for enhancing maize productivity. This study assessed newly developed maize hybrids under three planting densities on two sowing dates. The evaluated hybrids were 40 maize genotypes comprised of 36 F1-developed hybrids and 4 commercial high-yielding check hybrids. The developed hybrids were generated from selected maize inbred lines according to their adaptive traits to high planting density, such as prolificacy, erect leaves, short plants, early silking, anthesis-silking interval, and small tassel size. The applied planting densities were high, intermediate, and low, with 95,000, 75,000, and 55,000 plants/ha, respectively, under timely and late sowing. The high planting density displayed the uppermost grain yield compared with the intermediate and low densities at both sowing dates. The developed hybrid G36 exhibited the highest agronomic performance under high planting density at timely and late sowing. Additionally, G38, G16, G37, G23, G5, G31, G18, G7, G2, G20, G29, and G17 displayed high agronomic traits at both sowing dates. Joint regression and AMMI analyses revealed significant genotype, agro-environment, and genotype × agro-environment interaction effects for grain yield. The AMMI biplot displayed that G39 was closest to the ideal stable hybrid, and the hybrids G36, G18, G38, G17, G2, and G37 were considered desirable stable hybrids. Moreover, the GGE biplot indicated that a high planting density at an optimal sowing date could be considered a representative environment for discriminating high-yielding maize hybrids. The designated promising hybrids are recommended for further inclusion in maize breeding due to their stability and high yields.

## 1. Introduction

Maize (*Zea mays* L.) is an important cereal crop grown globally in different environments [1]. Maize occupies a prominent position in the agricultural sector worldwide; each part of the plant is utilized. It is a staple food, a source for animal feeding, and a bioenergy-producing crop. There is an urgent need to increase production, particularly under the fast-growing global population and abrupt global climate change [2]. Developing high-yielding and adopted hybrids is one of the best approaches to increasing maize production [3,4]. Its global acreage is around 202 million hectares, producing approximately 1163 million tonnes [5].

Improving maize production is subject to proper agricultural practices alongside genetically improved hybrids. Planting density is a vital yield component that considerably impacts maize production [6,7]. Modern maize hybrids are more cost-effective in utilizing nutrients and water and are tolerant to high planting density [8,9]. Optimal maize planting density varies considerably depending upon genetic properties, fertility status, maturity group, available water, and growing climatic conditions [10]. Conversely, extra higher planting density reduces maize grain yield because it could lead to lodging risk and increase the anthesis-silking interval and barrenness [11]. In this context, previous researchers depicted the response of modern maize hybrids to high planting density [12,13,14]. Accordingly, identifying the optimum planting density for newly developed maize hybrids is an irreplaceable management decision. Another decisive factor is the sowing date, which controls maize production, particularly under the current climate changes. Optimizing the sowing date for maize is essential to successfully producing a high grain yield [15]. Furthermore, identifying high-yielding and stable hybrids under diverse agro-environments is imperative for ameliorating maize production [16]. Maize grain yield is determined by genotype (G), environment (E), and genotype by environment interaction (GEI) effects. Each genotype reacts specifically to the environmental conditions and agronomic factors with high or low GEI [17]. The GEI refers to the numerous responses of genotypes through a wide range of environments [18,19]. The additive main effects and multiplicative interaction (AMMI) analysis is extensively performed due to its accuracy in graphically illustrating the complex interaction between genotypes and environments [20,21]. Similarly, the genotype plus genotype by environment interaction (GGE) biplot successfully analyzes genotype and environment interactions [22].

The aim of the present investigation was (i) to assess the performance of newly developed maize hybrids under three planting densities at timely and late sowing, (ii) to explore the impact of genotype, agro-environment, and their interaction on grain yield in different 40 maize hybrids; and (iii) to determine stable hybrids under different planting densities and sowing dates using joint regression, AMMI, and GGE analyses. The results of this study could provide helpful information in maize breeding.

## 2. Results

### 2.1. Analysis of Variance

The combined analysis of variance displayed that the main effects of sowing date, planting density, and genotype were significant for all measured traits (Table 1). The tested hybrids contributed 35.0% of the total variation in the sum of squares of grain yield, while sowing date and planting density explained 29.6% and 12.1% of the total variation. In addition, most of the agronomic traits were significantly influenced by the two-way and three-way interactions among sowing date, planting density, and genotype.

### 2.2. Mean Performance

The agronomic performance of the evaluated 36 hybrids and 4 commercial hybrid checks under 3 planting densities across two sowing dates are presented in Table 2, Table 3 and Table 4. All measured traits significantly varied among the tested maize hybrids across different agro-environments of sowing dates and plant densities. The timely sowing exhibited the highest performance for all traits. The optimum sowing date enhanced plant height, number of rows/ear, number of kernels/row, 100-kernel weight, and grain yield by 3.7, 1.4, 7.6, 8.3, and 22.4% compared to the late sowing date. Similarly, the high planting density enhanced plant height and grain yield by 5.1 and 16.3% compared to low density. Otherwise, high-density planting decreased the number of rows/ear, number of kernels/row, and 100-kernel weight by 3.7, 9.8, and 10.7%.

All maize hybrids demonstrated responses to sowing date and planting density with considerable variations in their performance. Plant height ranged from 208.2 to 314.7 cm; the highest values were assigned to the hybrids G36, G40, G37, G39, G10, and G11 under high planting density at timely sowing and G36, G40, G39, G38, G37, G10, and G3 at late sowing (Table 2). The number of rows/ear ranged from 12.2 to 16.7; the hybrids G20, G7, G38, G39, G36, G29, G25, and G26 exhibited the highest number of rows/ear under high planting density at timely sowing, while G18, G38, G25, G36, G26, G9, G17, G39 exhibited the highest number of rows/ear under late sowing (Table 2). The number of kernels/row varied from 29.7 to 45.9; the highest values were recorded by G36, G2, G38, G6, G39, G3, and G33 under high planting density at timely sowing and G36, G34, G40, G12, G31, G14 G22, and G39 at late sowing (Table 3). The 100-kernel weight ranged from 24.0 to 35.3 g; the hybrids G36, G8, G16, G37, G23, G5, G31, and G18 produced the highest kernel index under high planting density at timely sowing, while G16, G29, G7, G9, G39, G38, G2, G20, G37 and G2 produced the highest kernel index at late sowing (Table 3). Grain yield varied from 4.3 to 16.8 ton ha^−1^. The highest values were proved by G36, followed by G38, G16, G2, G39, G37, G20, G31, and G7 under high planting density at timely sowing and G36, G16, G39, G29, G17, G7, G38, G2, and G20 at late sowing (Table 4).

### 2.3. Genotypic Classification according to Agronomic Performance

The evaluated agronomic traits were employed to classify the assessed hybrids according to their performance under high planting density at timely and late sowing. Using hierarchical clustering, the hybrids were classified into five groups under high planting density at timely sowing (Figure 1a) and three groups at late sowing (Figure 1b). Group A included hybrids with the highest agronomic traits under high planting density at both sowing dates (Figure 1a,b). Likewise, group B included hybrids with intermediate performance, while those in groups C, D, and E had low agronomic performance compared to groups A and B.

### 2.4. Phenotypic Stability Parameters

Phenotypic stability parameters have been computed according to Eberhart and Russell [23] for the grain yield of the evaluated 40 maize hybrids. Joint regression analysis of variance showed that E + G × E and the agro-environment (linear) were highly significant, while G × E (linear) was not significant (Table 5). The importance of both linear (b_i_) and non-linear (s^2^_di_) sensitivity for the trait expression was evident. The results in Table 6 indicated that most evaluated hybrids exhibited a negative phenotypic index (Pi) except G2, G7, G9, G10, G14, G15, G16, G17, G18, G20, G24, G25, G26, G29, G36, G37, G38, G39, and G40. The regression coefficient (b_i_) of the maize hybrids ranged from 0.52 (G30) to 1.43 (G2), indicating the genetic variability among evaluated hybrids in their regression response (Table 6). The deviations from regression (s^2^_di_) ranged from 0.04 (G22 and G39) to 2.32 (G33). The b_i_ values were higher than the unity (bi > 1) and non-significant S^2^d_j_ for the hybrids G2, G5, G8, G14, G15, G16, G25, G31, G32, G34, G37, and G38. These hybrids are suitable for favorable agro-environments, including planting density, sowing date, and other inputs. Meanwhile, the b_i_ values were less than the unity (b_i_ < 1) with non-significant S^2^d_j_ in the hybrids G1, G3, G4, G7, G9, G18, G19, G21, G22, G24, G26, G27, G28, G29, and G30. Thus these hybrids could be grown under unfavorable conditions. Furthermore, the hybrids G6, G12, G13, G20, G23, G35, G36, G39, and G40 showed b_i_ values that were close to unity and with non-significant S^2^d_j_ (Table 6). Consequently, these hybrids are stable and suitable for overall agro-environments. According to Breese [24], genotypes with regression coefficients greater than unity could be adapted to favorable environments. Otherwise, those with a coefficient less than the unity could be relatively better adapted to less favorable growing conditions.

Taking into consideration the mean performance ($\bar{g}$), regression coefficient value (b_i_) and deviation from the regression (s^2^_di_), the most desirable and stable hybrid could be G36, which exhibited the highest grain yield, a regression coefficient value close to unity, and non-significant deviation from the regression. Similarly, G39, G18, and G20 displayed a mean performance above the grand mean, and their regression coefficients (b_i_) did not differ significantly from unity; also, minimum deviation mean squares (S^2^_di_) were detected. These hybrids could be helpful in maize breeding programs to improve grain yield under high planting densities at both sowing dates.

### 2.5. AMMI Analysis

The AMMI analysis showed that the maize hybrids (G), agro-environments (E), and their interaction (G × E) were significant for grain yield (Table 7). The proportion of sums of squares varied for genotypes (35.02%), environments (43.49%), and GEI (8.67%). The AMMI1 biplot showed that G36 had the highest mean yield, followed by G38, G39, G2, G7, G20, and G37. Among these hybrids, G39, G2, G38, and G36 had the lowest score of IPC1 (Figure 2a). Among the tested hybrids, G33 had the highest IPCA1 value of 1.07, while the lowest IPCA1 value (−0.82) was recorded by G16 (Figure 2, Table 7). Most hybrids showed a specific adaptation to the conditions of E1 (high planting density at timely sowing) and E2 (intermediate planting density at timely sowing) compared to the other agro-environments. The E1 presented the highest grain yield, with an IPC1 score close to zero indicating small interactions, while the highest IPCA1 value was assigned for E3 (low planting density at timely sowing). The environments E3 (low planting density at timely sowing), E4 (intermediate planting density at late sowing), and E6 (low planting density at late sowing) were unstable and more responsive as they were located far from the origin, whereas E2 and E5 were less responsive. Likewise, in AMMI2, the hybrids G18, G35, G36, G13, G26, and check hybrids G39 and G38 were more stable, as they were located near the origin (Figure 2b). On the contrary, the hybrids G33, G16, G32, G19, G24, and G12 were located far from the origin. Moreover, environments E1, E4, and E6 were unstable and more responsive as they were located far from the origin.

### 2.6. GGE Biplot

The angle among the environment vectors reveals their association. The agro-environments E1, E2, and E3 were positively correlated because their angles were smaller than 90°, as well as the environments E5, E6, and E4. The environments E3 and E4 were negatively associated (Figure 3a)**.** The ideal test environment was E1, with large IPCA1 and small IPCA2 scores. Moreover, E2 was a favorable environment, but E5 and E6 were unfavorable ones with environmental stress. GGE biplot graph for the SREG model is illustrated in Figure 3b. The maize hybrid G36 was an ideal maize hybrid with the highest vector length of high yield with zero GE. This indicates its high yield with the best performance across all agro-environments. Furthermore, the other hybrids G39, G38, G2, G37, and G17 are desirable and closer to the ideal hybrid.

## 3. Discussion

Breeding for high planting density is a decisive factor in increasing maize production [25,26]. In the present study, 36 newly developed maize hybrids and 4 commercial high-yielding hybrids were evaluated under 3 different plant densities at timely and late sowing. The developed hybrids were generated from diverse parental inbred lines assembled from different origins: CIMMYT, Egypt, Pioneer International Company (Johnston, Iowa, USA), and Thailand. Accordingly, the combined analysis of variance for grain yield and its attributes deduced highly significant genotypic differences among the evaluated maize hybrids. The evaluated hybrids contributed a higher proportion of the total variation for the yield traits compared to the sowing date and planting density. These findings were confirmed by the results of joint regression and AMMI analyses for grain yield across different agro-environments. The detected genetic divergences displayed the potential of used parental lines and their cross hybrids in improving maize genetic diversity and its potentiality for exploitation through maize breeding. Likewise, significant differences in maize agronomic traits regarding environmental and genotypic effects have also been stated by Acosta-Pech, et al. [27], Mafouasson, et al. [28], Al-Naggar, et al. [29], and Badu-Apraku, et al. [30].

Successful maize production depends on the adequate application of management inputs which increase and sustain agricultural production [31,32,33]. The agronomic performance of maize hybrids varied across tested agro-environments. The evaluated yield traits were largely influenced by the sowing date and planting density. Identifying the optimal sowing date is decisive for efficacious maize production, particularly under current climate change [34,35]. The sowing date on 10 April displayed the highest agronomic performance compared to the late date on 28 May. Sowing delay negatively impacted the plant height, the number of rows/ear, the number of kernels/row, 100-kernel weight, and the grain yield. Although late sowing decreased the grain yield by 18.3% compared to timely sowing, it is occasionally applied for intensive farming by cultivating different crops in the same land area annually in the Mediterranean region. Planting density is another vital factor for providing equal opportunity to the plants for survival and the best use of other inputs [36]. It depends on both intra-row spacing and row width. Low planting density reduces grain yield per unit, while extreme density leads to environmental stress on the plants [37]. Increasing planting density could decrease the productivity of the individual plant due to increasing the competition for available resources, but the grain yield per unit area increases. In the present study, increasing planting density up to 95,000 plant ha^−1^ distinctly increased the grain yield of all evaluated maize hybrids at both sowing dates. The significant enhancement in grain yield under high planting densities concurs with previous studies [38,39,40]. In this context, Yu, et al. [12] demonstrated a 7.5% elevated grain yield of maize hybrids by increasing planting density from 75,000 to 90,000 plants ha^−1^. Likewise, Assefa, et al. [6], Xu, et al. [10], and Al-Naggar, et al. [38], disclosed a significant boost in grain yield by increasing planting density up to 95,000 plants.

Determining the phenotypic variability of developed hybrids under different agro-environmental conditions could enable maize breeders to explore the genetic potential of these hybrids to improve maize productivity. The developed hybrids displayed different responses to the evaluated agro-environmental conditions. Generally, the hybrid G36, followed by G38, G16, G37, G23, G5, G31, G18, G7, G2, G20, G29, and G17, recorded the highest grain yield attributes under different agro-environments compared to the other hybrids. These hybrids displayed good phenotypic adaptation to different plant densities and sowing dates. The detected phenotypic adaptation of agronomic performance is ascribed to their genetic architecture, which is valuable for future maize breeding [41,42,43,44]. On the other hand, G35 and G21 presented the lowest phenotypic adaptation across all tested environments.

The agro-environmental effect and its interaction with genotype are considered significant challenges facing maize breeders to identify superior hybrids for wide adaptation. Genotype by environment (G × E) interaction is critical in developing new maize hybrids across different environments. Significant G × E interaction indicates that the hybrids are not stable across different environments [28]. Maize breeders should develop stable hybrids across different environments to be widely accepted by farmers. In this study, the evaluated agro-environment (E) had the largest contribution to the total variation of grain yield (43.49%), followed by genotypic effect (35.02%) and G × E interaction (8.67%). This tendency of components (E ˃ G ˃ G × E) has been previously demonstrated by Bocianowski, et al. [45], Al-Naggar, et al. [46], and Abate [47]. The genotype × planting density interaction had a highly significant effect on yield traits as well as sowing date × planting density interaction and other interactions involving planting density. Similarly, Al-Naggar, et al. [46] and Ajayo, et al. [48] elucidated significant genotype interactions by agronomic factors on yield traits in maize.

The significant E + (G × E) component for grain yield indicated that the hybrids responded differently in the tested agro-environments. Furthermore, the mean squares of environments (linear) were highly significant, indicating the considerable differences among the studied agro-environments. Earlier researchers depicted significant differences among maize hybrids evaluated in diverse environments [49,50]. Eberhart and Russell [23] highlighted the importance of linear and non-linear components of G × E interaction, b_i_ and S^2^_di_, in arbitrating the phenotypic stability of the genotypes. In this model, b_i_ is considered a parameter of response, and its values close to the unity reveal less responsive to environmental change and more adaptation. Otherwise, negative b_i_ values indicate that the hybrid may be grown only in a poor environment [17]. According to this model, maize hybrids G36, G39, G18, and G26 were more stable than the other genotypes, while maize hybrids G7, G17, and G29 had high-yielding unstable hybrids. On the other hand, the hybrid G35 was among the lowest yielding hybrids in this study but was stable in the tested agro-environments. In general, the stable developed hybrids that exhibited high-yielding qualities, such as G36, should be further evaluated in multi-environment trials and could be exploited for increasing commercial maize production. Correspondingly, Badu-Apraku, et al. [30] and Sserumaga, et al. [51] applied the model of Eberhart and Russell [23] for identifying stable maize hybrids evaluated in different environments.

AMMI and GGE biplot analyses could assist maize breeders in making better decisions for identifying stable and adapted hybrids [51]. An AMMI biplot provides a good visual assessment of G × E based on grain yield. It revealed large variability between tested maize hybrids and agro-environments. The AMMI biplot illustrated that the hybrids G18, G26, G36, and G39 were more stable and high-yielding. The vertex hybrids were G16, G2, G15, G33, G30, and G24. The vertex hybrid in each sector represented the highest yielding genotype in the environment that fell within that particular sector [50,52]. Furthermore, a GGE biplot was applied to identify the best hybrids in each mega-environment and assess the stability of tested hybrids [20]. The polygon vertices of the GGE biplot were G33, G38, G36, G16, G19, and G35; these hybrids were located extremely far from the origin in different directions. These hybrids were more responsive than those within the polygon. The scatter biplot was divided into six sectors and two mega-environments. The first mega-environment consisted of E2 (intermediate planting density at timely sowing), and E3 (low planting density at timely sowing) had G38 as the high-yielding hybrid. The second mega-environment, comprised or E1, E2, E4, E5, and E6, had G36 as the highest yielding hybrid. No environment fell within the sector with G35, G33, G32, G16, and G19, indicating that these genotypes were the poorest hybrids in all tested agro-environments or were not the greatest in any of the mega-environments.

## 4. Materials and Methods 

### 4.1. Plant Material and Developing F1 Hybrids

Nine yellow maize inbred lines in the self seventh-generation isolated from different sources were used in this study. According to the origin of used parental inbred lines, one was from CIMMYT, five were from Egypt, one was from Pioneer International Company (Johnston, IA, USA), and two were from Thailand (Table 8). The inbred lines were selected from preliminary screening trials according to their adaptive traits to high planting density, such as prolificacy, erect leaves, short plants, early silking, anthesis-silking interval, and small tassel size.

In the summer season of 2017, the nine S_7_ inbred lines were sown at the Experimental Stations of Fine Seeds International, Beba district, Beni Suef Governorate, Egypt (28°54′06.5″ N, 30°56′21.2″ E) on two sowing dates (May 21st and on June 6th) to obtain enough quantities of hybrid seeds. Each inbred line was sown in 20 5-m long rows which were 0.70 m wide. A total of 2 seeds were sown per hill spaced 20 cm apart along the row. The plants were thinned to one plant per hill before the first irrigation. At the flowering time, all possible cross combinations (without reciprocals) were made among the nine inbred lines (half-diallel); consequently, 36 F_1_ hybrids were obtained.

### 4.2. Field Trail

The field experiment was conducted in the summer of 2018 at Al Fant, Al-Fashn district, Beni-Suef Governorate (28°45′02.4″ N, 30°52′25.9″ E). The preceding crop was Egyptian clover/Berseem (*Trifolium alexandrinum* L.). The experimental soil was classified as clay soil (21.58% sand, 22.61% silt, and 55.82% clay) which was slightly alkaline. The soil properties are illustrated in Table 9. The experimental site had a hot desert climate, and the two growing seasons were extremely hot and dry. The maximum temperature was close to 40 °C, and precipitation was absent most of the two seasons (Figure 4). The Nile River was the main water source for irrigation. Extreme temperature and humidity conditions can affect dehiscence. A decrease in the pollen grain shed and viability can also reduce the fertilization of an ovule. Therefore, the used parental inbred lines were produced and tested previously under regional climate conditions. The experimental site is in full sun from April to October; it basks in over 4000 h of annual sunshine. June was the longest month in the year with a day length average of 14 h, which then decreased to 13.7, 13.1, 12.3, and 11.5 in July, August, September, and October.

The evaluated maize hybrids were comprised of 36 diallel crosses and four high-yielding commercial yellow maize single-cross hybrids: SC-176, Pioneer-32D99, Fine-276, and Fine-354. The tested hybrids were evaluated under three planting densities and two different sowing dates. The two sowing dates were timely on 10 April and late on 28 May. The three planting densities were high planting density (HPD: 95,000 plants ha^−1^), intermediate planting density (MPD: 75,000 plants ha^−1^), and low planting density (LPD: 55,000 plants ha^−1^). A split-plot design in an alpha lattice (5 × 8) arrangement was used with three replications. The main plots were assigned to three planting densities on each sowing date, and the sub-plots were allocated to maize hybrids. The sub-plot area was 7 m^2^, and it consisted of 2 rows, 5 m long and 0.70 m wide, with a distance of 15 cm, 19 cm, and 26 cm between hills, for high, intermediate, and low planting densities, respectively. Trials at six agro-environments were hand-planted with two seeds per hill, and thinning to one plant per hill was carried out three weeks after planting and before the first irrigation. Phosphorus, potassium, and nitrogen fertilizers were applied at rates of 75 kg P_2_O_5_ ha^−1^, 115 kg K_2_O, and 300 kg N ha^−1^. The other recommended agricultural practices for maize cultivation were applied properly throughout the growing season. The harvest dates were 5 August in the first sowing and 12 September in the second season.

### 4.3. Measured Traits

The plant height, number of rows/ear, number of kernels/row, 100-kernel weight, and grain yield/plant were measured for the evaluated hybrids. At physiological maturity, the plant height was recorded as the distance from the soil surface to the tassel base for 10 plants at each plot. Twenty ears were harvested randomly from each plot to measure the number of rows and grains per ear. All separated ears of each plot were shelled and then weighed for grain yield calculated as kg/ha.

### 4.4. Statistical Procedures

The combined analysis of variance was performed for grain yield and its attributes using Genstat statistical software [53]. Genotype and environment were considered fixed effects, but replication was considered a random effect. Bartlett’s test used the homogeneity of variance to compare the within-environment error mean squares (MS) before proceeding to the stability analysis using the method outlined by Gomez and Gomez [54]. The phenotypic index (P_i_) was estimated according to Ram, et al. [55]. The phenotypic stability parameters, deviation from regression (S^2^di), and regression coefficient (bi) were determined following Eberhart and Russell [23]. Significant differences between b_i_ value and unity were confirmed by *t*-tests among S^2^d_i_ and zero-tested by F-test. The AMMI model was estimated according to Gauch [20] and Crossa, et al. [56]. The AMMI stability value (ASV) was calculated according to Purchase, et al. [57]. The AMMI and SREG (site regression) models obtained the GE biplot and GGE biplot, respectively. Biplots of the first two principal components (PCA) were performed to display these relationships among the evaluated hybrids and environments [58].

## 5. Conclusions

Highly significant genetic divergences were distinguished among the evaluated 40 maize hybrids. The detected genetic variations illustrated the potential of used parental lines and their cross hybrids in improving maize genetic diversity, which could be exploited through maize breeding. The evaluated yield traits were largely influenced by the sowing date and planting density. Although late sowing declined grain yield by 18.3% compared to timely sowing, it is occasionally applied for intensive farming. Increasing the planting density increased the grain yield of all evaluated maize hybrids at both sowing dates. The maize hybrids displayed different responses to the evaluated agro-environmental conditions. Generally, the hybrid G36, followed by G39, G38, G37, G2, G7, G20, and G29, recorded the highest grain yield and attributes under different agro-environments compared to the other hybrids. Joint regression, AMMI, and GGE analyses could be employed to identify stable maize hybrids in diverse environments. The stability analyses illustrated that the hybrids G39, G36, G18, G38, G17, G2, and G37 are desirable stable hybrids. Accordingly, these hybrids are recommended for further inclusion in maize breeding to improve maize production.

## Figures and Tables

**Figure 1 plants-11-01187-f001:**
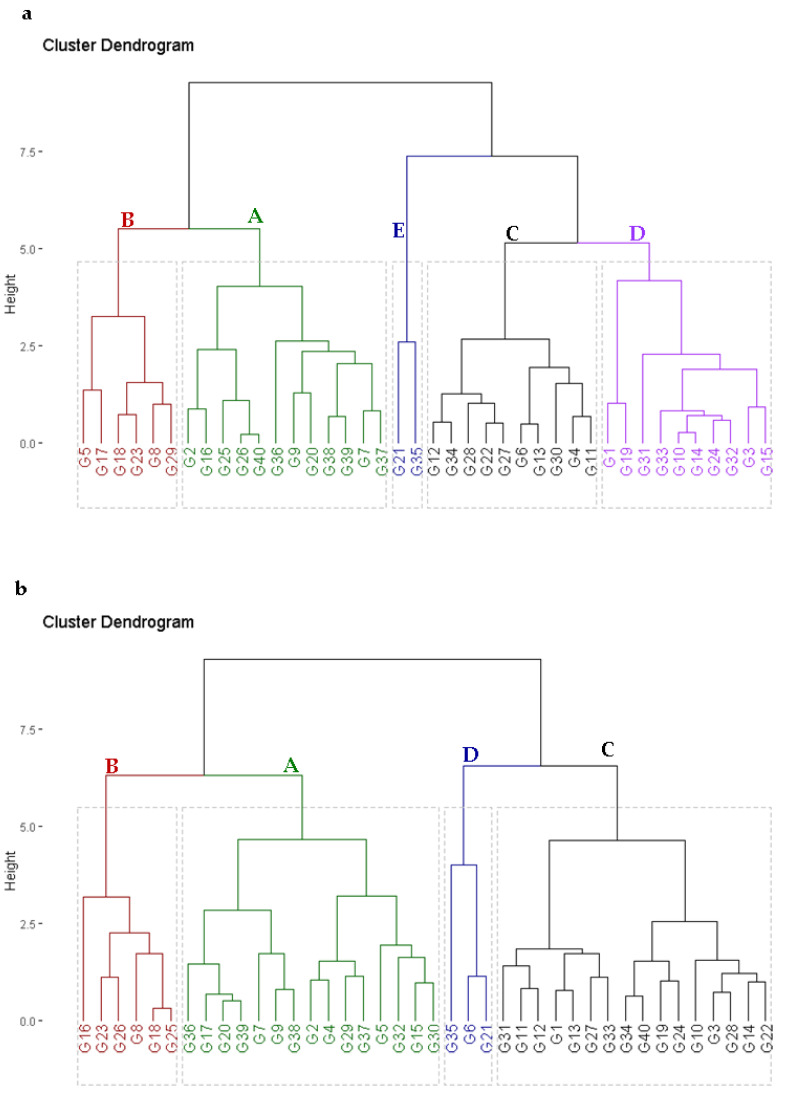
Dendrogram of the phenotypic distances among 40 maize hybrids based on their agronomic performance under high planting density at timely sowing (**a**) and late sowing (**b**).

**Figure 2 plants-11-01187-f002:**
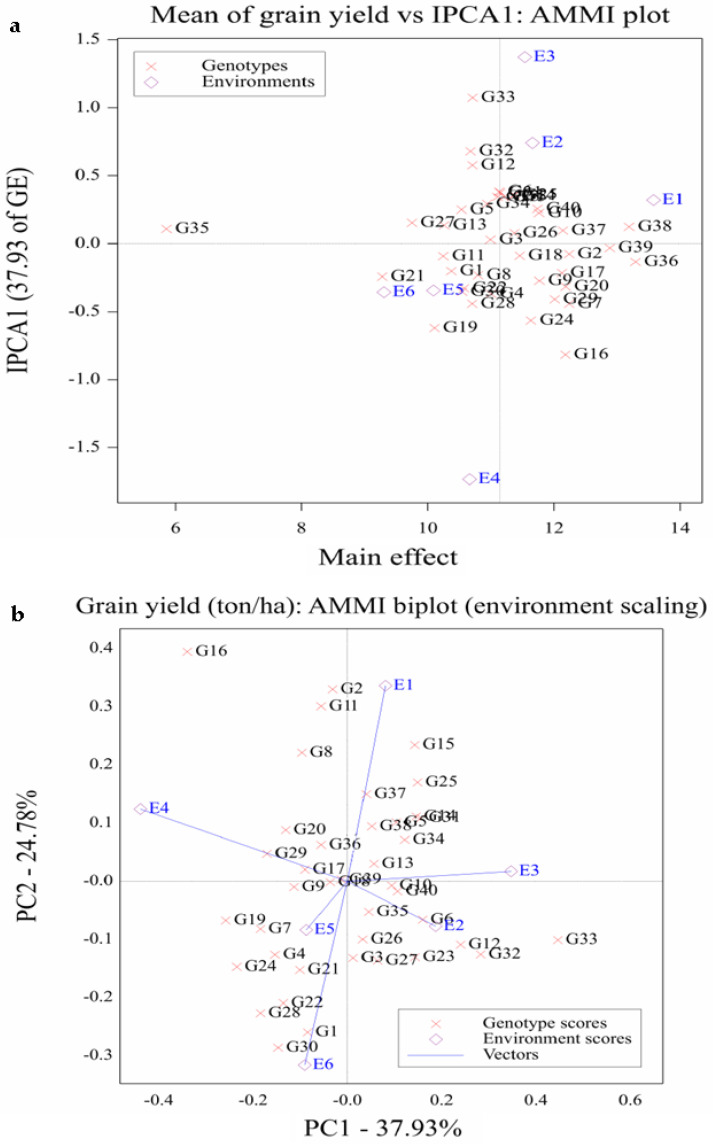
AMMI1 (**a**) and AMMI2 (**b**) biplots for grain yield of 40 developed maize hybrids and 4 commercial checks (assessed as G1–G40) evaluated in six agro-environments (coded as E1–E6).

**Figure 3 plants-11-01187-f003:**
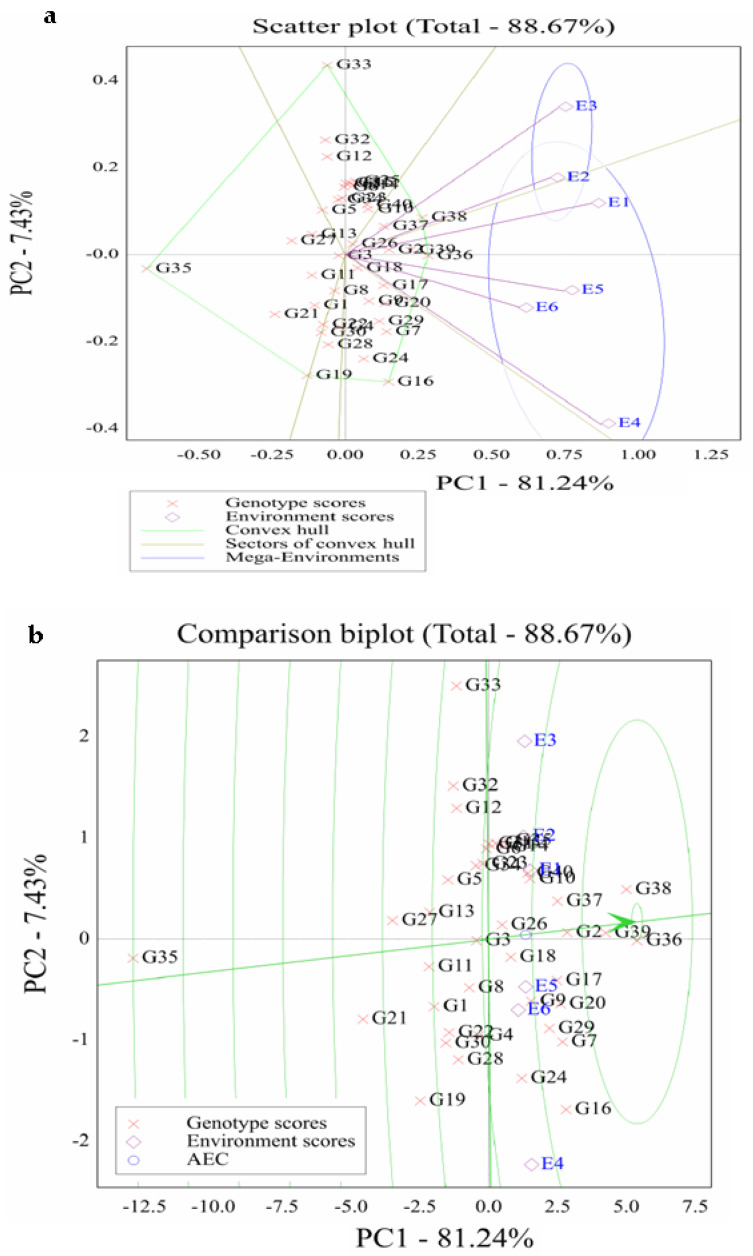
Scatter plot (**a**) and comparison graphic (**b**) of GGE biplot for grain yield of 40 maize hybrids (assessed G1–G40) across six agro-environments (assessed E1–E6). The green arrow directed tested hybrids toward greater grain yield, and the other vertical green line crosses biplot origin displays the stability.

**Figure 4 plants-11-01187-f004:**
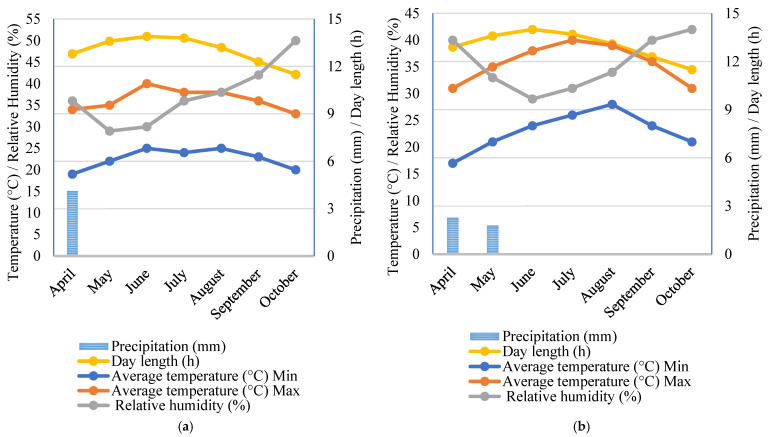
Monthly air temperature (°C), relative humidity (%), precipitation (mm), and day length (h) in the first (**a**) and second growing seasons (**b**).

**Table 1 plants-11-01187-t001:** Combined analyses of variance (mean square) of 40 hybrids under three planting densities in two sowing dates for grain yield and its attributes.

Source of Variation	df	Plant Height	Number of Rows/Ear	Number of Kernels/Row	100-Kernel Weight	Grain Yield (ton ha^−1^)
Sowing date (SD)	1	849.77 **	8.02 *	1407.28 **	371.23 **	903.69 **
Planting density (PD)	2	9791.23 **	4.20 *	74.79 **	345.68 **	185.56 **
SD × PD	2	6033.45 **	2.53 *	75.13 **	121.58 **	27.2 *
Error a	12	1312.62	1.62	10.07	2.68	5.92
Genotypes (G)	39	4247.60 **	9.59 **	64.80 **	7.61 **	27.44 **
G × SD	39	238.65 **	0.94 *	6.11	4.65	2.17 **
G × PD	78	98.61	0.76	4.69	6.78 **	1.41 **
G × SD × PD	78	124.54	0.56	5.97	7.85 **	0.90 *
Error b	468	112.67	0.59	5.48	3.84	0.69
Total	719	408.00	1.16	11.11	6.61	4.25

df: degree of freedom; *: significant at *p* < 0.05; **: significant at *p* < 0.01.

**Table 2 plants-11-01187-t002:** Impact of sowing date and planting density on plant height and the number of rows/ear of the evaluated 40 maize hybrids.

Hybrid	Plant Height (cm)							Number of Rows/Ear						
	Timely Sowing			Late Sowing			Mean	Timely Sowing			Late Sowing			Mean
	HPD	MPD	LPD	HPD	MPD	LPD		HPD	MPD	LPD	HPD	MPD	LPD	
G1 (L1 × L2)	297.0	286.0	278.3	285.3	278.5	274.3	283.2	13.1	13.3	13.9	12.9	13.5	13.7	13.4
G2 (L1 × L3)	281.3	281.0	278.7	283.7	272.7	264.6	277.0	14.3	14.9	15.3	13.7	14.5	15.2	14.7
G3 (L1 × L4)	281.0	272.3	271.3	286.3	265.7	262.0	273.1	14.3	15.3	15.3	14.4	15.5	14.9	15.0
G4 (L1 × L5)	283.7	273.3	272.3	271.3	268.7	261.4	271.8	13.3	12.9	14.4	13.1	12.9	13.7	13.4
G5 (L1 × L6)	273.3	281.7	273.0	282.0	269.7	258.5	273.0	14.4	14.7	14.1	14.1	14.9	14.5	14.5
G6 (L1 × L7)	274.0	272.7	277.0	276.0	251.7	254.2	267.6	13.6	13.5	14.3	13.7	13.1	13.5	13.6
G7 (L1 × L8)	285.7	282.0	275.7	280.0	267.0	262.1	275.4	15.5	14.9	15.5	14.4	14.8	14.8	15.0
G8 (L1 × L9)	284.0	273.0	263.7	275.0	263.3	255.1	269.0	14.3	14.8	14.3	14.5	14.8	14.8	14.6
G9 (L2 × L3)	272.3	251.7	264.3	256.0	254.3	246.9	257.6	14.4	15.2	15.1	15.1	14.9	14.3	14.8
G10 (L2 × L4)	303.3	285.0	288.3	290.0	280.0	275.4	287.0	14.0	14.8	15.3	14.7	15.2	14.9	14.8
G11 (L2 × L5)	298.0	280.7	274.3	276.3	281.7	267.4	279.7	13.6	13.6	13.5	13.1	13.3	12.3	13.2
G12 (L2 × L6)	279.7	279.7	281.7	282.0	280.3	265.6	278.2	14.0	13.5	14.7	13.6	14.4	13.6	14.0
G13 (L2 × L7)	288.7	279.7	281.0	281.0	279.7	265.6	279.3	13.3	13.6	14.8	12.5	12.9	13.1	13.4
G14 (L2 × L8)	276.3	274.0	276.3	279.0	272.7	260.1	273.1	14.0	15.1	15.7	14.0	14.0	14.5	14.6
G15 (L2 × L9)	292.3	279.0	268.1	280.0	276.0	266.7	277.0	14.4	13.9	14.9	14.0	14.1	14.1	14.2
G16 (L3 × L4)	295.7	274.0	264.3	272.7	263.0	259.3	271.5	14.9	15.3	14.5	15.3	16.0	15.1	15.2
G17 (L3 × L5)	284.3	275.3	268.0	266.3	280.3	259.1	272.2	14.7	14.8	14.9	14.9	14.4	15.3	14.8
G18 (L3 × L6)	282.0	273.7	276.7	270.7	259.7	258.9	270.3	14.4	14.5	15.5	15.6	15.2	16.1	15.2
G19 (L3 × L7)	276.0	257.0	247.7	254.0	262.0	241.5	256.4	13.6	14.0	14.8	13.7	14.1	14.7	14.2
G20 (L3 × L8)	262.3	262.3	271.0	251.3	259.0	245.4	258.5	16.0	16.3	16.7	14.7	15.2	16.3	15.9
G21 (L3 × L9)	252.3	254.7	229.0	236.7	245.7	228.4	241.1	13.9	15.5	15.2	14.0	14.4	14.5	14.6
G22 (L4 × L5)	293.7	275.7	256.0	270.3	274.3	259.6	271.6	14.0	14.4	14.7	14.8	14.0	14.7	14.4
G23 (L4 × L6)	284.3	282.3	277.7	275.0	260.7	261.1	273.5	13.9	12.8	14.1	14.7	14.3	14.8	14.1
G24 (L4 × L7)	296.0	276.0	276.0	267.0	264.7	261.4	273.5	13.7	14.9	15.3	13.7	14.3	14.5	14.4
G25 (L4 × L8)	274.7	270.0	263.0	259.3	249.3	248.5	260.8	14.8	15.5	16.1	15.5	16.1	15.6	15.6
G26 (L4 × L9)	279.0	272.3	261.0	268.7	274.3	254.3	268.3	14.8	15.1	14.9	15.2	14.4	15.3	15.0
G27 (L5 × L6)	283.0	261.7	269.0	265.0	267.0	252.1	266.3	13.9	14.0	14.4	13.3	14.1	13.9	13.9
G28 (L5 × L7)	278.0	264.3	258.3	248.0	263.3	247.9	260.0	14.4	13.7	14.7	14.1	13.6	13.3	14.0
G29 (L5 × L8)	269.3	265.3	250.0	262.0	262.0	248.9	259.6	14.9	13.9	14.7	13.3	14.9	14.0	14.3
G30 (L5 × L9)	281.7	269.3	258.0	261.7	272.0	253.8	266.1	13.2	14.0	14.5	13.6	13.6	13.7	13.8
G31 (L6 × L7)	266.0	257.3	250.7	252.3	253.3	241.7	253.6	13.1	12.9	14.0	13.2	13.5	12.9	13.3
G32 (L6 × L8)	260.0	261.7	259.0	256.3	255.7	241.8	255.7	14.0	14.9	16.4	14.0	15.3	14.5	14.9
G33 (L6 × L9)	268.0	253.0	257.3	255.0	255.7	242.4	255.2	14.1	14.4	14.9	13.7	13.9	14.0	14.2
G34 (L7 × L8)	265.7	265.7	259.0	257.7	234.7	239.1	253.6	14.3	14.9	14.5	13.9	13.3	14.1	14.2
G35 (L7 × L9)	222.0	231.7	222.7	228.0	224.7	208.2	222.9	12.5	13.2	13.3	12.2	14.1	14.0	13.2
G36 (L8 × L9)	314.7	296.3	305.0	305.4	287.1	288.8	299.5	14.1	14.9	14.7	13.3	13.3	14.4	14.1
G37 (SC-176)	306.3	283.3	281.7	291.0	283.3	274.9	286.8	14.5	14.9	15.3	14.1	15.1	15.6	14.9
G38 (Pioneer-32D99)	288.0	285.0	284.7	291.7	274.0	269.1	282.1	15.3	15.7	15.9	15.6	16.1	16.3	15.8
G39 (Fine-276)	303.7	300.7	303.0	294.8	290.4	286.7	296.5	15.3	16.0	16.2	14.9	16.3	15.9	15.8
G40 (Fine-354)	312.3	301.0	301.0	300.7	281.7	283.7	296.7	14.8	14.9	14.0	14.0	14.4	14.0	14.4
Mean	281.7	273.0	269.3	271.1	266.5	257.4		14.2	14.5	14.9	14.1	14.4	14.5	
Mean SD			274.7			265.0				14.5			14.3	
RLSD_0.05_ (SD)	11.7							0.11						
RLSD_0.05_ (PD)	12.4							0.14						
RLSD_0.05_ (G)	13.3							0.18						
RLSD_0.05_ (G × SD × PD)	18.3							0.31						

HPD: high planting density (95,000 plants ha^−1^); MPD: intermediate planting density (75,000 plants ha^−1^); LPD: low planting density (55,000 plants ha^−1^).

**Table 3 plants-11-01187-t003:** Impact of sowing date and planting density on the number of kernels/row and 100-kernel weight of the evaluated 40 maize hybrids.

Hybrid	Number of Kernels Per Row							100-Kernel Weight (g)						
	Timely Sowing			Late Sowing			Mean	Timely Sowing			Late Sowing			Mean
	HPD	MPD	LPD	HPD	MPD	LPD		HPD	MPD	LPD	HPD	MPD	LPD	
G1 (L1 × L2)	38.1	40.5	42.5	36.5	38.1	39.9	39.3	26.5	29.0	30.7	26.0	30.3	31.4	29.0
G2 (L1 × L3)	41.3	41.1	44.5	34.0	36.0	39.9	39.5	29.4	33.7	33.9	28.0	28.0	32.0	30.8
G3 (L1 × L4)	41.0	42.3	43.9	36.5	36.8	39.7	40.0	28.0	31.0	33.0	25.7	30.7	32.0	30.1
G4 (L1 × L5)	39.8	42.8	40.2	34.8	37.1	37.7	38.7	30.4	31.7	32.3	27.7	30.3	31.7	30.7
G5 (L1 × L6)	34.7	36.7	38.4	33.2	33.1	36.6	35.5	32.1	32.3	33.7	27.7	29.0	32.3	31.2
G6 (L1 × L7)	38.9	43.0	43.6	32.2	35.9	39.1	38.8	32.0	31.7	32.3	25.7	28.0	30.7	30.1
G7 (L1 × L8)	39.8	41.3	40.6	33.7	36.5	41.7	38.9	30.3	30.0	31.7	29.3	28.4	31.1	30.1
G8 (L1 × L9)	39.2	42.3	44.2	32.1	36.5	39.9	39.0	32.7	29.0	32.3	26.0	28.3	33.0	30.2
G9 (L2 × L3)	39.1	40.4	41.1	35.5	37.9	40.8	39.1	31.0	30.7	32.5	28.7	30.0	32.4	30.9
G10 (L2 × L4)	40.1	42.3	43.1	36.6	41.1	41.9	40.9	29.7	30.0	32.3	24.0	30.0	30.3	29.4
G11 (L2 × L5)	40.7	42.0	44.7	37.6	40.2	42.3	41.3	30.3	31.5	34.2	25.0	28.7	31.3	30.2
G12 (L2 × L6)	40.1	42.2	45.1	38.5	41.5	41.1	41.4	30.7	32.0	31.7	25.0	29.0	29.3	29.6
G13 (L2 × L7)	36.7	38.1	41.3	37.6	39.7	41.3	39.1	29.0	31.0	34.7	26.3	27.4	33.1	30.3
G14 (L2 × L8)	40.4	40.6	41.5	37.8	39.3	42.1	40.3	29.4	31.7	31.7	25.0	27.3	30.3	29.2
G15 (L2 × L9)	40.7	44.1	45.8	36.5	39.9	41.3	41.4	28.4	32.7	31.5	26.7	29.1	33.2	30.3
G16 (L3 × L4)	37.1	40.2	42.3	29.7	33.2	43.1	37.6	32.3	34.0	34.7	29.7	28.3	32.3	31.9
G17 (L3 × L5)	34.1	36.6	41.3	36.4	36.4	38.4	37.2	30.2	30.0	31.7	27.3	29.3	31.3	30.0
G18 (L3 × L6)	37.3	40.4	42.6	33.2	36.3	40.5	38.4	32.0	28.3	32.3	25.3	26.0	30.7	29.1
G19 (L3 × L7)	36.7	38.0	40.9	37.1	34.7	35.7	37.2	27.1	29.3	33.1	26.7	29.3	31.3	29.5
G20 (L3 × L8)	38.3	37.6	40.7	37.0	35.8	35.5	37.5	29.7	31.0	32.0	28.0	28.3	31.4	30.1
G21 (L3 × L9)	32.8	35.1	38.8	31.8	33.1	32.6	34.0	31.0	33.0	33.4	26.7	28.0	29.3	30.2
G22 (L4 × L5)	39.4	41.0	41.6	37.7	37.9	39.5	39.5	31.0	32.3	33.3	25.3	28.7	29.3	30.0
G23 (L4 × L6)	37.7	41.0	40.8	36.0	36.7	38.7	38.5	32.3	31.7	33.3	27.0	26.0	32.7	30.5
G24 (L4 × L7)	39.9	42.6	44.4	36.7	39.5	41.6	40.8	29.0	29.3	33.3	26.0	25.3	29.3	28.7
G25 (L4 × L8)	37.7	41.3	42.9	33.8	37.5	41.5	39.1	28.3	28.7	32.0	25.3	28.3	30.0	28.8
G26 (L4 × L9)	38.9	44.0	44.7	34.5	39.2	42.1	40.6	29.3	30.7	32.0	26.3	30.0	28.7	29.5
G27 (L5 × L6)	39.7	39.9	41.5	36.4	37.3	39.7	39.1	30.3	31.3	32.0	26.7	28.7	31.7	30.1
G28 (L5 × L7)	39.5	40.4	43.2	35.4	39.7	41.9	40.0	29.7	31.0	31.0	25.3	29.7	29.3	29.3
G29 (L5 × L8)	39.5	41.2	44.9	36.8	37.4	37.7	39.6	32.0	30.7	32.7	29.7	30.0	30.7	31.0
G30 (L5 × L9)	38.7	41.0	43.5	35.5	37.8	40.4	39.5	31.3	31.0	33.0	27.7	27.7	30.7	30.2
G31 (L6 × L7)	41.2	40.0	43.0	38.1	42.7	42.5	41.3	28.3	30.0	32.7	26.3	28.7	30.3	29.4
G32 (L6 × L8)	39.2	41.4	44.8	35.9	38.2	39.5	39.8	29.3	31.3	35.3	27.7	31.0	32.4	31.2
G33 (L6 × L9)	40.9	43.4	44.4	36.8	42.1	42.4	41.7	29.3	30.0	32.0	25.3	31.0	31.3	29.8
G34 (L7 × L8)	40.3	40.7	41.6	39.3	39.9	39.6	40.2	30.3	32.3	34.3	26.3	27.4	30.0	30.1
G35 (L7 × L9)	33.5	36.6	37.7	31.7	36.7	36.3	35.4	29.3	30.0	33.0	27.3	27.0	29.7	29.4
G36 (L8 × L9)	42.1	44.7	45.9	39.8	41.1	42.5	42.7	32.7	33.0	33.0	26.7	29.3	31.1	31.0
G37 (SC-176)	39.7	41.3	44.9	36.4	39.9	40.2	40.4	32.3	30.3	31.0	28.0	30.3	33.2	30.9
G38 (Pioneer-32D99)	41.3	42.8	45.3	34.7	40.5	39.9	40.8	30.0	29.7	34.1	28.7	30.0	33.0	30.9
G39 (Fine-276)	41.1	43.6	44.2	37.7	42.5	43.2	42.1	30.3	31.4	33.2	28.7	28.3	31.7	30.6
G40 (Fine-354)	39.4	43.3	44.0	38.9	40.4	42.3	41.4	29.3	31.3	33.0	26.0	31.0	31.7	30.4
Mean	38.9	40.9	42.8	35.8	38.2	40.1		30.2	31.0	32.7	26.8	28.8	31.2	
Mean SD			40.9			38.0				31.3			28.9	
RLSD_0.05_ (SD)	1.4							1.8						
RLSD_0.05_ (PD)	0.7							2.1						
RLSD_0.05_ (G)	1.6							2.4						
RLSD_0.05_ G × SD × PD	2.8							3.0						

HPD: high planting density (95,000 plants ha^−1^); MPD: intermediate planting density (75,000 plants ha^−1^); LPD: low planting density (55,000 plants ha^−1^).

**Table 4 plants-11-01187-t004:** Impact of sowing date and planting density on grain yield (ton ha^−1^) of the evaluated 40 maize hybrids.

Hybrid	Timely Sowing			Late Sowing			Mean
	HPD	MPD	LPD	HPD	MPD	LPD	
G1 (L1 × L2)	12.05	11.15	10.04	9.66	10.11	9.22	10.37
G2 (L1 × L3)	15.78	13.27	11.94	12.14	11.15	9.17	12.24
G3 (L1 × L4)	13.18	11.55	11.46	10.37	9.52	9.90	11.00
G4 (L1 × L5)	12.88	11.16	11.05	11.06	10.15	9.72	11.00
G5 (L1 × L6)	13.78	10.97	11.14	9.58	8.94	8.81	10.53
G6 (L1 × L7)	13.35	11.76	12.16	9.81	10.64	9.07	11.13
G7 (L1 × L8)	14.39	12.77	11.77	12.39	11.28	10.87	12.24
G8 (L1 × L9)	13.80	10.47	11.53	11.29	8.94	8.78	10.80
G9 (L2 × L3)	14.34	12.19	11.46	11.53	10.93	10.17	11.77
G10 (L2 × L4)	13.57	12.98	13.10	11.74	9.51	9.69	11.77
G11 (L2 × L5)	13.42	9.71	10.63	9.64	9.18	8.90	10.25
G12 (L2 × L6)	13.07	11.74	11.78	9.02	9.63	9.02	10.71
G13 (L2 × L7)	13.04	10.76	10.56	9.23	9.78	8.20	10.26
G14 (L2 × L8)	13.79	11.93	12.93	10.92	9.54	8.97	11.35
G15 (L2 × L9)	14.09	11.70	12.59	10.72	9.87	8.27	11.21
G16 (L3 × L4)	15.86	12.14	10.90	13.19	11.74	9.26	12.18
G17 (L3 × L5)	13.84	12.65	12.71	12.43	11.58	9.55	12.13
G18 (L3 × L6)	13.55	11.50	12.19	11.22	10.96	9.34	11.46
G19 (L3 × L7)	12.17	10.26	9.62	10.64	9.20	8.77	10.11
G20 (L3 × L8)	15.19	12.85	11.55	12.11	11.03	10.40	12.19
G21 (L3 × L9)	10.90	10.47	9.18	9.32	8.07	7.74	9.28
G22 (L4 × L5)	12.05	11.03	10.68	10.58	9.75	9.37	10.58
G23 (L4 × L6)	13.57	12.07	11.66	9.74	9.68	9.90	11.11
G24 (L4 × L7)	13.22	11.82	11.40	12.05	11.11	10.22	11.64
G25 (L4 × L8)	13.79	12.22	12.90	11.04	9.87	8.43	11.38
G26 (L4 × L9)	13.94	11.58	11.85	10.46	10.04	10.38	11.37
G27 (L5 × L6)	12.23	10.04	10.10	8.40	9.18	8.53	9.75
G28 (L5 × L7)	11.95	11.26	10.56	10.85	10.34	9.27	10.71
G29 (L5 × L8)	14.21	12.62	11.81	12.45	11.16	9.83	12.01
G30 (L5 × L9)	11.86	10.80	10.66	10.44	9.79	9.71	10.54
G31 (L6 × L7)	14.51	11.66	11.49	9.48	10.92	8.82	11.15
G32 (L6 × L8)	13.52	11.54	11.58	8.44	9.51	9.47	10.68
G33 (L6 × L9)	12.89	13.17	11.92	8.15	10.22	7.94	10.71
G34 (L7 × L8)	13.37	11.94	11.98	10.40	9.38	8.60	10.94
G35 (L7 × L9)	8.52	6.43	6.23	5.03	4.33	4.61	5.86
G36 (L8 × L9)	16.83	13.39	13.46	13.28	12.05	10.76	13.29
G37 (SC-176)	15.22	12.26	12.83	11.60	10.93	10.02	12.14
G38 (Pioneer-32D99)	16.31	13.19	13.81	12.38	12.06	11.42	13.19
G39 (Fine-276)	15.34	13.10	13.45	12.51	11.73	11.20	12.89
G40 (Fine-354)	13.90	12.43	13.00	11.26	9.90	9.95	11.74
Mean	13.58	11.66	11.54	10.66	10.09	9.31	
Mean SD			12.26			10.02	
RLSD_0.05_ (SD)	1.13						
RLSD_0.05_ (PD)	1.06						
RLSD_0.05_ (G)	1.27						
RLSD_0.05_ G × SD × PD	2.43						

HPD: high planting density (95,000 plants ha^−1^); MPD: intermediate planting density (75,000 plants ha^−1^); LPD: low planting density (55,000 plants ha^−1^).

**Table 5 plants-11-01187-t005:** Joint regression analysis of variance for grain yield of 40 hybrids across six agro-environments.

Source of Variation	df	Sum of Squares	Mean Square
Model	239	888.19	98.22 **
Genotype (G)	39	356.76	9.15 **
Environment (E)	5	443.07	88.61 **
G × E	195	88.36	0.45 **
E + G × E	200	531.43	2.66 **
Environment (linear)	1	443.07	443.07 **
G × E (linear)	39	23.04	0.59 ^ns^
Pooled deviation	160	65.31	0.41 **
Pooled error	468	106.92	0.23

^ns^: not significant; **: significant at *p* < 0.01; df: degree of freedom.

**Table 6 plants-11-01187-t006:** Stability parameters of the evaluated 40 maize hybrids for grain yield (ton ha^−1^) under two sowing dates and three plant densities.

Genotypes	$\mathrm{Mean}\;({\bar{g}}_{i})$	P_i_	b_i_	S^2^_di_	IPCA_1_	IPCA_2_	ASV	Rank
G1 (L1 × L2)	10.37	−0.77	0.64 *	0.22	−0.20	0.56	0.64	26
G2 (L1 × L3)	12.24	1.10	1.43 **	0.39	−0.08	−0.71	0.72	31
G3 (L1 × L4)	11.00	−0.15	0.87	0.16	0.03	0.29	0.29	8
G4 (L1 × L5)	11.00	−0.14	0.71 **	0.07	−0.37	0.27	0.63	24
G5 (L1 × L6)	10.53	−0.61	1.23	0.16	0.25	−0.22	0.44	15
G6 (L1 × L7)	11.13	−0.01	1.00	0.38	0.38	0.14	0.61	22
G7 (L1 × L8)	12.24	1.10	0.79	0.24	−0.44	0.18	0.70	30
G8 (L1 × L9)	10.80	−0.34	1.16	0.62 *	−0.23	−0.48	0.60	20
G9 (L2 × L3)	11.77	0.63	0.93	0.16	−0.27	0.02	0.42	14
G10 (L2 × L4)	11.77	0.62	1.06	0.86 **	0.23	0.02	0.35	10
G11 (L2 × L5)	10.25	−0.89	1.33 *	0.48 *	−0.13	−0.65	0.68	29
G12 (L2 × L6)	10.71	−0.43	1.06	0.54 *	0.58	0.24	0.91	35
G13 (L2 × L7)	10.26	−0.88	1.08	0.18	0.14	−0.06	0.22	5
G14 (L2 × L8)	11.35	0.21	1.20	0.45	0.35	−0.24	0.59	19
G15 (L2 × L9)	11.21	0.06	1.34 *	0.29	0.34	−0.51	0.73	32
G16 (L3 × L4)	12.18	1.04	1.24	1.94 **	−0.82	−0.85	1.51	39
G17 (L3 × L5)	12.13	0.99	0.89	0.46 *	−0.22	−0.04	0.33	9
G18 (L3 × L6)	11.46	0.32	0.90	0.20	−0.09	0.00	0.13	2
G19 (L3 × L7)	10.11	−1.03	0.73	0.38	−0.62	0.15	0.96	37
G20 (L3 × L8)	12.19	1.05	1.08	0.38	−0.32	−0.19	0.52	18
G21 (L3 × L9)	9.28	−1.86	0.78	0.30	−0.24	0.33	0.50	17
G22 (L4 × L5)	10.58	−0.57	0.63 **	0.04	−0.33	0.45	0.68	28
G23 (L4 × L6)	11.11	−0.04	1.00	0.39	0.34	0.28	0.60	21
G24 (L4 × L7)	11.64	0.50	0.62 **	0.21	−0.56	0.32	0.92	36
G25 (L4 × L8)	11.38	0.23	1.27	0.50 *	0.36	−0.37	0.66	27
G26 (L4 × L9)	11.37	0.23	0.92	0.26	0.08	0.22	0.25	6
G27 (L5 × L6)	9.75	−1.39	0.88	0.32	0.15	0.30	0.38	12
G28 (L5 × L7)	10.71	-0.44	0.56 **	0.15	−0.44	0.49	0.84	34
G29 (L5 × L8)	12.01	0.87	0.93	0.34	−0.41	−0.10	0.63	25
G30 (L5 × L9)	10.54	−0.60	0.52 **	0.02	−0.35	0.62	0.82	33
G31 (L6 × L7)	11.15	0.00	1.26	0.62 *	0.38	−0.24	0.62	23
G32 (L6 × L8)	10.68	−0.46	1.11	0.98 **	0.68	0.27	1.08	38
G33 (L6 × L9)	10.71	−0.43	1.26	2.32 **	1.07	0.22	1.66	40
G34 (L7 × L8)	10.94	−0.20	1.18	0.20	0.29	−0.15	0.47	16
G35 (L7 × L9)	5.86	−5.28	1.01	0.19	0.11	0.11	0.20	3
G36 (L8 × L9)	13.29	2.15	1.04	0.25	−0.09	−0.15	0.20	4
G37 (SC 176)	12.14	1.00	1.20 *	0.08	0.10	−0.32	0.36	11
G38 (SC Pioneer 32D99)	13.19	2.05	1.14	0.17	0.12	−0.20	0.28	7
G39 (Fine 276)	12.89	1.75	0.98	0.04	−0.03	−0.01	0.05	1
G40 (Fine 354)	11.74	0.60	1.05	0.36	0.26	0.04	0.39	13

*: significant at *p* < 0.05; **: significant at *p* < 0.01; ${\bar{g}}_{i}$: mean of genotype; ($P_{i}$): phenotypic index; b_i_: regression coefficient; S^2^_di_: mean square deviations from linear regression; ASV: AMMI stability value.

**Table 7 plants-11-01187-t007:** AMMI analysis of grain yield (ton ha^−1^) variance of 40 maize hybrids across six agro-environments.

Source of Variation	Df	Sum of Squares	Mean Squares	Percent
Environments (E)	5	1329.20	265.84 **	43.49
Rep (Envi.)	12	71.00	5.92 **	8.64
Genotypes (G)	39	1070.30	27.44 **	35.02
G × E	195	265.10	1.36 **	8.67
IPCA1	43	100.50	2.34 **	37.93
IPCA2	41	65.70	1.60 **	24.78
Residuals	111	98.90	0.89 *	37.31
Pooled Error	468	320.80	0.69	
Total	719	3056.40		

df: degree of freedom; Envi.: environment; Rep: replication; IPCA: interaction principal component axis; *: significant at *p* < 0.05; **: significant at *p* < 0.01.

**Table 8 plants-11-01187-t008:** Designation, origin, and place of 9 inbred lines (L) that were used in this study.

No.	Entry Designation	Origin	Institution	Flowering Date		Height (cm)		Prolific	Grain Yield
				Anth.	Silk.	Plant	Eear	%	(ton ha^−1^)
L1	CML 114	Population-45	CIMMYT *	60	63	133	72	25	4.87
L2	L249	FSI	Fine Seeds Int.	61	63	195	80	0	4.13
L3	4883	Ajeeb	Fine Seeds Int.	57	59	110	62	40	4.53
L4	4893	Shams	Misr Hytech	58	60	113	70	0	3.23
L5	5166	72013	Golden Seed	57	59	123	55	20	5.28
L6	YL13-M 0325	155/30N11	Pion. Hi-Bred Int.	59	59	132	55	30	5.45
L7	YL14-A 0407	S.C. 164	Agric. Res. Center	56	58	148	72	10	4.08
L8	YL14-A 0444	Population-1	Agric. Res. Fund. Agency	57	58	133	67	20	5.45
L9	YL15-M 0534	Mon. C599	Agric. Res. Fund. Agency	57	60	117	50	1	5.40

* CIMMYT: International Maize and Wheat Improvement Center, México-Veracruz, Mexico; Fine Seeds Int.: Fine Seeds International, Giza, Egypt; Misr Hytech: Misr Hytech Seed, Cairo, Egypt; Golden Seed: Golden West Seed, Beni-Suef, Egypt; Pion. Hi-Bred Int.: Pioneer Hi-Bred International, Johnston, IA, USA; Agric. Res. Center: Agricultural Research Center, Giza, Egypt; Agric. Res. Fund. Agency: Agricultural Research Funding Agency, Bangkok, Thailand.

**Table 9 plants-11-01187-t009:** Soil physical and chemical analyses of the experimental site.

Depth (cm)	Sand (%)	Silt (%)	Clay (%)	Texture	Organic Matter(g kg^−1^)	pH	EC (dS m^−1^)	CaCO_3_ (g kg^−1^)	Available N (mg kg^−1^)	Available P (mg kg^−1^)	Available K (mg kg^−1^)
0.0–30	20.18	23.56	56.26	Clay	15.4	7.83	1.22	22.2	9.98	20.6	179.7
31–60	22.97	21.65	55.38	Clay	13.9	7.96	1.38	23.8	8.5	16.1	122.1

## Data Availability

The data presented in this study are available upon request from the corresponding author. The data are not publicly available due to privacy of research participants.

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
