# Peer review of "Multivariate Analysis of Agronomic Traits in Newly Developed Maize Hybrids Grown under Different Agro-Environments"

_plants, 2022, doi:10.3390/plants11091187_

Round 1
Reviewer 1 Report
Among the environmental factors, are there any factors that determine whether corn has diseases and insect pests?
Author Response
Reviewer 1:
Among the environmental factors, are there any factors that determine whether corn has diseases and insect pests?
Re: We would like to thank the Reviewer for his time devoted to our manuscript. The used parental inbred lines for generating the 36 F1 hybrids were selected according to their adaptive traits to high planting density and resistance to late wilt disease, which is common in Egypt. There were no detected symptoms of late wilt disease during the evaluation of the developed hybrids under different planting densities at timely and late sowing. Moreover, in the breeding program, all identified promising high-yielding hybrids from different crosses will be assessed in an infected nursery under artificial soil inoculation by the pathogen Magnaporthiopsis maydis.
Reviewer 2 Report
The MS aimed to assess newly developed maize hybrids include 36 F1 developed hybrids and 4 commercial high-yielding check hybrids under three planting densities in two sowing dates.
1, The MS can briefly introduce the origin, relationship or difference between these 40 maizes.
2, The MS analyzed the phenotype of 40 maize varieties in the experiment in detail, and compared the differences by statistical analysis, but why the differences of phenotype produced about 40 maize varieties. For example, why can g36, 40, 39, 37, 10 adapt to both high planting density at timly sowing and late sowing?
Author Response
Reviewer 2:
The MS aimed to assess newly developed maize hybrids including 36 F1 developed hybrids and 4 commercial high-yielding check hybrids under three planting densities in two sowing dates.
Re: We would like to thank the Reviewer for his time devoted to our manuscript.
1, The MS can briefly introduce the origin, relationship or difference between these 40 maizes.
Re: Thanks for your suggestion, this point has been elucidated in the discussion section, please see lines 229-238 and 268-273.
2, The MS analyzed the phenotype of 40 maize varieties in the experiment in detail, and compared the differences by statistical analysis, but why the differences of phenotype produced about 40 maize varieties. For example, why can G36, 40, 39, 37, 10 adapt to both high planting density at timely sowing and late sowing?
Re: Thanks for your suggestion; more explanations on phenotypic adaptation and its relation to genetic architecture have been clarified in the discussion section (lines 266-276)
Round 2
Reviewer 2 Report
The MS had been revised according to the comments of reviewer, So I no other comments. And The MS can be accepted for the Journal.